# Temporal Variability in Implicit Online Learning

**Nicolò Campolongo**[*]
Università di Milano
nicolo.campolongo@unimi.it

**Francesco Orabona**
Boston University
francesco@orabona.com

## Abstract

In the setting of online learning, Implicit algorithms turn out to be highly successful from a practical standpoint. However, the tightest regret analyses only show marginal improvements over Online Mirror Descent. In this work, we shed light on this behavior carrying out a careful regret analysis. We prove a novel static regret bound that depends on the temporal variability of the sequence of loss functions, a quantity which is often encountered when considering dynamic competitors. We show, for example, that the regret can be constant if the temporal variability is constant and the learning rate is tuned appropriately, without the need of smooth losses. Moreover, we present an adaptive algorithm that achieves this regret bound without prior knowledge of the temporal variability and prove a matching lower bound. Finally, we validate our theoretical findings on classification and regression datasets.

## 1 Introduction

The *online learning* paradigm is a powerful tool to model common scenarios in the real world when the data comes in a streaming fashion, for example in the case of time series. In the last two decades there has been a tremendous amount of progress in this field (see, e.g., [30, 13, 24], for an introduction), which also led to advances in seemingly unrelated areas of machine learning and computer science. In this setting, a learning agent faces the environment in a game played sequentially. The protocol is the following: given a time horizon $T$, in every round $t = 1, \ldots, T$ the agent chooses a model $\boldsymbol{x}_t$ from a convex set $V$. Then, a convex loss function $\ell_t$ is revealed by the environment and the agent pays a loss $\ell_t(\boldsymbol{x}_t)$. As usual in this setting, we do not make assumptions about the environment, but allow it to be adversarial. The agent's goal is to minimize her regret against any decision maker, i.e., the cumulative sum of her losses compared to the losses of an agent which always commits to the same choice $\boldsymbol{u}$. So, formally the regret against any $\boldsymbol{u} \in V$ is defined as

$$R_T(\boldsymbol{u}) \triangleq \sum_{t=1}^{T} \ell_t(\boldsymbol{x}_t) - \sum_{t=1}^{T} \ell_t(\boldsymbol{u}) \ .$$

Much of the progress in this field is driven by the strictly related model of Online Linear Optimization (OLO): exploiting the assumption that the loss functions are convex, we can linearize them using a first-order approximation through its (sub)gradient and subsequently minimize the linearized regret. For example, the well-known Online Gradient Descent (OGD) [38] simply uses the direction of the negative (sub)gradient of the loss function to update its model, multiplied by a given learning rate. Usually, a properly tuned learning rate gives a regret bound of $\mathcal{O}(\sqrt{T})$, which is also optimal. On the other hand, we can choose to not use any approximation to the loss function and instead update our model using directly the loss function rather than its subgradient [17]. This type of update is known as *Implicit* and algorithms designed in this way are known to have practical advantages [18]. Unfortunately, their theoretical understanding is still limited at this point.

---

[*]Work done while visiting the OPTIMAL Lab at Boston University.

Our first contribution (Section 5) in this paper is a refined analysis of Implicit algorithms in the framework of Online Mirror Descent (OMD). Doing this allows us to understand why Implicit algorithms might practically work better compared to algorithms which use (sub)gradients in the update. In particular, we describe how these algorithms can potentially incur only a constant regret if the sequence of loss functions does not vary with time. In particular, we measure the hardness of the sequence of loss functions with its *temporal variability*, which is defined as

$$V_T \triangleq \sum_{t=2}^{T} \max_{\boldsymbol{x} \in V} \ \ell_t(\boldsymbol{x}) - \ell_{t-1}(\boldsymbol{x}) \ . \tag{1}$$

Our second contribution (Section 6) is a new adaptive Implicit algorithm, AdaImplicit, which retains the worst-case $\mathcal{O}(\sqrt{T})$ regret bound but takes advantage of a slow varying sequence of loss functions and achieve a regret of $\mathcal{O}(V_T + 1)$. Also, we prove a lower bound which shows that our algorithm is optimal. Finally, in order to show the benefits of using Implicit algorithms in practice, in Section 7 we conduct an empirical analysis on real-world datasets in both classification and regression tasks.

## 2 Related Work

**Implicit Updates.** The implicit updates in online learning were proposed for the first time by Kivinen and Warmuth [17]. However, such update with the Euclidean divergence is the Proximal update in the optimization literature dating back at least to 1965 [22, 19, 29, 27], and more recently used even in the stochastic setting [33, 2]. Later, this idea was re-invented by Crammer et al. [11] for the specific case of linear prediction with losses that have a range of values in which they are zero, e.g., hinge loss and epsilon-insensitive loss. Implicit updates were also used for online learning with kernels [9] and to deal with importance weights [16]. Kulis and Bartlett [18] provide the first regret bounds for implicit updates that match those of OMD, while McMahan [20] makes the first attempt to quantify the advantage of the implicit updates in the regret bound. Finally, Song et al. [31] generalize the results in McMahan [20] to Bregman divergences and strongly convex functions, and quantify the gain differently in the regret bound. Note that in [20, 31] the gain cannot be exactly quantified, providing just a non-negative data-dependent quantity subtracted to the regret bound.

**Adaptivity.** Our new analysis hinges on the concept of *temporal variability* $V_T$ of the losses, a quantity first defined in Besbes et al. [5] in the context of non-stationary stochastic optimization and later generalized in Chen et al. [8]. In general, the temporal variability has been used in works considering dynamic environments [e.g., 15, 37, 3, 36]. In particular, Jadbabaie et al. [15] consider different notions of adaptivity at the same time: if we consider the static regret case with no optimistic updates, then their bound gives $R_T = \tilde{\mathcal{O}}(\sqrt{\sum_{t=1}^{T} \|\boldsymbol{g}_t\|_\star^2 + 1})$, which is never better than ours. At first sight, our algorithm seems to achieve the same constant regret bound of Optimistic algorithms [10, 28] if the sequence of loss functions is such that $V_T = \mathcal{O}(1)$. However, for this result Optimistic algorithms need either smooth or linear loss functions. In contrast, our algorithm does not need this assumption. Other examples of adaptivity to the sequence of loss functions can be found in [14, 32], which consider bounds in terms of the variance of the sequence of linear losses.

Finally, it is worth mentioning that recently there have been attempts to analyze Implicit algorithms in dynamic environments [see, e.g., 12, 1, 7]. Nevertheless, these works are not directly comparable to ours since they either consider a different (noisy) setting and competitor or make stronger assumptions (i.e. smoothness and/or strong convexity of the loss functions).

## 3 Definitions

For a function $f : \mathbb{R}^d \to (-\infty, +\infty]$, we define a *subgradient* of $f$ in $\boldsymbol{x} \in \mathbb{R}^d$ as a vector $\boldsymbol{g} \in \mathbb{R}^d$ that satisfies $f(\boldsymbol{y}) \geq f(\boldsymbol{x}) + \langle \boldsymbol{g}, \boldsymbol{y} - \boldsymbol{x} \rangle, \ \forall \boldsymbol{y} \in \mathbb{R}^d$. We denote the set of subgradients of $f$ in $\boldsymbol{x}$ by $\partial f(\boldsymbol{x})$. The *indicator function of the set* $V$, $i_V : \mathbb{R}^d \to (-\infty, +\infty]$, is defined as

$$i_V(\boldsymbol{x}) = \begin{cases} 0, & \boldsymbol{x} \in V, \\ +\infty, & \text{otherwise.} \end{cases}$$

We denote the *dual norm* of $\| \cdot \|$ by $\| \cdot \|_\star$. A proper function $f : \mathbb{R}^d \to (-\infty, +\infty]$ is $\mu$-*strongly convex* over a convex set $V \subseteq \text{int dom} f$ w.r.t. $\| \cdot \|$ if $\forall \boldsymbol{x}, \boldsymbol{y} \in V$ and $\boldsymbol{g} \in \partial f(\boldsymbol{x})$, we have

---

**Algorithm 1** Implicit Online Mirror Descent (IOMD)

---

**Require:** Non-empty closed convex set $V \subset X \subset \mathbb{R}^d$, $\psi : X \to \mathbb{R}$, $\eta_t > 0$, $\boldsymbol{x}_1 \in V$
1: **for** $t = 1, \ldots, T$ **do**
2:    Output $\boldsymbol{x}_t \in V$
3:    Receive $\ell_t : \mathbb{R}^d \to \mathbb{R}$ and pay $\ell_t(\boldsymbol{x}_t)$
4:    Update $\boldsymbol{x}_{t+1} = \arg\min_{\boldsymbol{x} \in V} \ B_\psi(\boldsymbol{x}, \boldsymbol{x}_t) + \eta_t \ell_t(\boldsymbol{x})$
5: **end for**

---

$f(\boldsymbol{y}) \geq f(\boldsymbol{x}) + \langle \boldsymbol{g}, \boldsymbol{y} - \boldsymbol{x} \rangle + \frac{\mu}{2} \|\boldsymbol{x} - \boldsymbol{y}\|^2$. Let $\psi : X \to \mathbb{R}$ be strictly convex and continuously differentiable on $\text{int } X$. The *Bregman Divergence* w.r.t. $\psi$ is $B_\psi : X \times \text{int } X \to \mathbb{R}_+$ defined as $B_\psi(\boldsymbol{x}, \boldsymbol{y}) = \psi(\boldsymbol{x}) - \psi(\boldsymbol{y}) - \langle \nabla \psi(\boldsymbol{y}), \boldsymbol{x} - \boldsymbol{y} \rangle$. We assume that $\psi$ is strongly convex w.r.t. a norm $\|\cdot\|$ in $\text{int } X$. We also assume w.l.o.g. the strong convexity constant to be 1, which implies

$$B_\psi(\boldsymbol{x}, \boldsymbol{y}) \geq \frac{1}{2}\|\boldsymbol{x} - \boldsymbol{y}\|^2, \quad \forall \boldsymbol{x} \in X, \boldsymbol{y} \in \text{int } X \ . \tag{2}$$

# 4    Online Mirror Descent with Implicit Updates

In this section, we introduce the Implicit Online Mirror Descent (IOMD) algorithm, its relationship with OMD, and some of its properties.

Consider a set $V \subset X \subseteq \mathbb{R}^d$. The Online Mirror Descent [35, 4] update over $V$ is

$$\boldsymbol{x}_{t+1} = \arg\min_{\boldsymbol{x} \in V} \ B_\psi(\boldsymbol{x}, \boldsymbol{x}_t) + \eta_t(\ell_t(\boldsymbol{x}_t) + \langle \boldsymbol{g}_t, \boldsymbol{x} - \boldsymbol{x}_t \rangle) = \arg\min_{\boldsymbol{x} \in V} \ B_\psi(\boldsymbol{x}, \boldsymbol{x}_t) + \eta_t \langle \boldsymbol{g}_t, \boldsymbol{x} \rangle,$$

for $\boldsymbol{g}_t \in \partial \ell_t(\boldsymbol{x}_t)$ received as feedback. In words, OMD updates the solution minimizing a first-order approximation of the received loss, $\ell_t$, around the predicted point, $\boldsymbol{x}_t$, constrained to be not too far from the predicted point measured with the Bregman divergence. It is well-known, [e.g. 24], that the regret guarantee for OMD for a non-increasing sequence of learning rates $(\eta_t)_{t=1}^T$ is

$$R_T(\boldsymbol{u}) \leq \sum_{t=1}^T \frac{B_\psi(\boldsymbol{u}, \boldsymbol{x}_t) - B_\psi(\boldsymbol{u}, \boldsymbol{x}_{t+1})}{\eta_t} + \sum_{t=1}^T \frac{\eta_t}{2}\|\boldsymbol{g}_t\|_\star^2, \quad \forall \boldsymbol{u} \in V \ . \tag{3}$$

This gives a $\mathcal{O}(\sqrt{T})$ regret with, e.g., $\max_{\boldsymbol{x}, \boldsymbol{y} \in V} B_\psi(\boldsymbol{x}, \boldsymbol{y}) < \infty$, Lipschitz losses, and $\eta_t \propto 1/\sqrt{t}$.

A natural variation of the classic OMD update is to use the actual loss function $\ell_t$, rather than its first-order approximation. This is called *implicit update* [17] and is defined as

$$\boldsymbol{x}_{t+1} = \arg\min_{\boldsymbol{x} \in V} \ B_\psi(\boldsymbol{x}, \boldsymbol{x}_t) + \eta_t \ell_t(\boldsymbol{x}) \ . \tag{4}$$

Note that, in general, this update does not have a closed form, but for many interesting cases it is still possible to efficiently compute it. Notably, for $\psi = \frac{1}{2}\|\cdot\|_2^2$ and linear prediction with the square, absolute, and hinge loss, these updates can all be computed in closed form when $V = \mathbb{R}^d$ [see, e.g., 11, 18]. This update leads to the Implicit Online Mirror Descent (IOMD) algorithm in Algorithm 1.

We next show how the update in Eq. (4) yields new interesting properties which are not shared with its non-implicit counterpart. Their proofs can be found in Appendix B.

**Proposition 4.1.** *Let $\boldsymbol{x}_{t+1}$ be defined as in Eq.* (4). *Then, there exists $\boldsymbol{g}_t' \in \partial \ell_t(\boldsymbol{x}_{t+1})$ such that*

$$\ell_t(\boldsymbol{x}_t) - \ell_t(\boldsymbol{x}_{t+1}) - B_\psi(\boldsymbol{x}_{t+1}, \boldsymbol{x}_t)/\eta_t \geq 0, \tag{5}$$

$$\langle \eta_t \boldsymbol{g}_t' + \nabla\psi(\boldsymbol{x}_{t+1}) - \nabla\psi(\boldsymbol{x}_t), \boldsymbol{u} - \boldsymbol{x}_{t+1} \rangle \geq 0 \quad \forall \boldsymbol{u} \in V, \tag{6}$$

$$\langle \boldsymbol{g}_t', \boldsymbol{x}_{t+1} - \boldsymbol{x}_t \rangle \geq \langle \boldsymbol{g}_t, \boldsymbol{x}_{t+1} - \boldsymbol{x}_t \rangle \ . \tag{7}$$

The first property implies that, in contrast to OMD, the value of the loss function in $\boldsymbol{x}_{t+1}$ is always smaller than or equal to its value in $\boldsymbol{x}_t$. This means that, if $\ell_t = \ell$, the value $\ell(\boldsymbol{x}_t)$ will be monotonically decreasing over time. The second property gives an alternative way to write the update rule expressed in Eq. (4). In particular, using $\psi(\boldsymbol{x}) = \frac{1}{2}\|\boldsymbol{x}\|_2^2$ and $V = \mathbb{R}^d$ the update becomes $\boldsymbol{x}_{t+1} = \boldsymbol{x}_t - \eta_t \boldsymbol{g}_t'$, motivating the name "implicit". Using this fact in the last property,[2] we have that with $L_2$ regularization, the dual norm of $\boldsymbol{g}_t'$ is smaller than the dual norm of $\boldsymbol{g}_t$, i.e. $\|\boldsymbol{g}_t'\|_2 \leq \|\boldsymbol{g}_t\|_2$.

Let's gain some additional intuition on the implicit updates. Consider the case of $V = \mathbb{R}^d$ and $\psi(\boldsymbol{x}) = \frac{1}{2}\|\cdot\|_2^2$. We have that $\boldsymbol{x}_{t+1} = \boldsymbol{x}_t - \eta_t \boldsymbol{g}_t'$, where $\boldsymbol{g}_t' \in \partial \ell_t(\boldsymbol{x}_{t+1})$. Now, if $\ell_{t+1} \approx \ell_t$, we would be updating the algorithm approximately with the *next subgradient*. On the other hand, knowing future gradients is a safe way to have constant regret. Hence, we can expect IOMD to have low regret if the functions are slowly varying over time. In the next sections, we will see that this is indeed the case.

## 5 Two Regret Bounds for IOMD

In the following, we will present a new regret guarantee for IOMD. First, we give a simple lemma that provides a bound on the cumulative losses paid *after* the updates (proof in Appendix B).

**Lemma 5.1.** *Let $V \subset X \subset \mathbb{R}^d$ be a non-empty closed convex set. Let $B_\psi$ be the Bregman divergence w.r.t. $\psi : X \to \mathbb{R}$. Then, Algorithm 1 guarantees*

$$\sum_{t=1}^{T} \ell_t(\boldsymbol{x}_{t+1}) - \sum_{t=1}^{T} \ell_t(\boldsymbol{u}) \leq \sum_{t=1}^{T} \frac{B_\psi(\boldsymbol{u}, \boldsymbol{x}_t) - B_\psi(\boldsymbol{u}, \boldsymbol{x}_{t+1})}{\eta_t} - \sum_{t=1}^{T} \frac{1}{\eta_t} B_\psi(\boldsymbol{x}_{t+1}, \boldsymbol{x}_t) . \quad (8)$$

*Furthermore, assume that $(\eta_t)_{t=1}^T$ is a non-increasing sequence and let $D^2 \triangleq \max_{\boldsymbol{x}, \boldsymbol{u} \in V} B_\psi(\boldsymbol{u}, \boldsymbol{x})$. Then the bound can further be expressed as*

$$\sum_{t=1}^{T} \ell_t(\boldsymbol{x}_{t+1}) - \sum_{t=1}^{T} \ell_t(\boldsymbol{u}) \leq \frac{D^2}{\eta_T} - \sum_{t=1}^{T} \frac{1}{\eta_t} B_\psi(\boldsymbol{x}_{t+1}, \boldsymbol{x}_t) . \quad (9)$$

Adding $\sum_{t=1}^{T} \ell_t(\boldsymbol{x}_t)$ on both sides of Eq. (8), we immediately get our new regret bound.

**Theorem 5.2.** *Under the assumptions of Lemma 5.1, the regret incurred by Algorithm 1 is bounded as*

$$R_T(\boldsymbol{u}) \leq \sum_{t=1}^{T} \frac{B_\psi(\boldsymbol{u}, \boldsymbol{x}_t) - B_\psi(\boldsymbol{u}, \boldsymbol{x}_{t+1})}{\eta_t} + \sum_{t=1}^{T} \left[ \ell_t(\boldsymbol{x}_t) - \ell_t(\boldsymbol{x}_{t+1}) - \frac{B_\psi(\boldsymbol{x}_{t+1}, \boldsymbol{x}_t)}{\eta_t} \right] . \quad (10)$$

We note that this result could also be extrapolated from [31], by carefully going through the proof of their Lemma 1. However, as in the other previous work, they did not identify that the key quantity to be used in order to quantify an actual gain is the temporal variability $V_T$, as we will show later.

**First Regret: Recovering OMD's Guarantee.** To this point, the advantages of an implicit update are still not clear. Therefore, we now show how, from Theorem 5.2, one can get a possibly tighter bound than the usual $\mathcal{O}(\sqrt{T})$. The key point in this new analysis is to introduce $\boldsymbol{g}_t'$ as defined in Proposition 4.1 and relate it to the Bregman divergence between $\boldsymbol{x}_t$ and $\boldsymbol{x}_{t+1}$.

**Theorem 5.3.** *Let $\boldsymbol{g}_t' \in \partial \ell_t(\boldsymbol{x}_{t+1})$ satisfy Eq. (6). Assume $\psi$ to be 1-strongly convex w.r.t. $\|\cdot\|$. Then, under the assumptions of Lemma 5.1, we have that Algorithm 1 satisfies*

$$\ell_t(\boldsymbol{x}_t) - \ell_t(\boldsymbol{x}_{t+1}) - \frac{B_\psi(\boldsymbol{x}_{t+1}, \boldsymbol{x}_t)}{\eta_t} \leq \eta_t \|\boldsymbol{g}_t\|_\star \min\left( 2\|\boldsymbol{g}_t'\|_\star, \frac{\|\boldsymbol{g}_t\|_\star}{2} \right), \forall t, \boldsymbol{g}_t \in \partial \ell_t(\boldsymbol{x}_t). \quad (11)$$

*Proof.* Using the convexity of the losses, we can bound the difference between $\ell_t(\boldsymbol{x}_t)$ and $\ell_t(\boldsymbol{x}_{t+1})$:

$$\ell_t(\boldsymbol{x}_t) - \ell_t(\boldsymbol{x}_{t+1}) \leq \langle \boldsymbol{g}_t, \boldsymbol{x}_t - \boldsymbol{x}_{t+1} \rangle \leq \|\boldsymbol{g}_t\|_* \|\boldsymbol{x}_t - \boldsymbol{x}_{t+1}\|,$$

where $\boldsymbol{g}_t \in \partial \ell_t(\boldsymbol{x}_t)$. Given that $\psi$ is 1-strongly convex, we can use Eq. (2) to obtain

$$\ell_t(\boldsymbol{x}_t) - \ell_t(\boldsymbol{x}_{t+1}) \leq \|\boldsymbol{g}_t\|_\star \sqrt{2 B_\psi(\boldsymbol{x}_{t+1}, \boldsymbol{x}_t)} . \quad (12)$$

Note that $\ell_t(\boldsymbol{x}_t) - \ell_t(\boldsymbol{x}_{t+1}) - B_\psi(\boldsymbol{x}_{t+1}, x_t)/\eta_t \leq \ell_t(\boldsymbol{x}_t) - \ell_t(\boldsymbol{x}_{t+1})$. Hence, to get the first term in the $\min$ of Eq. (11), we can simply look for an upper bound on the term $\sqrt{2 B_\psi(\boldsymbol{x}_{t+1}, \boldsymbol{x}_t)}$ in Eq. (12) above. Using the fact that the Bregman divergence is convex in its first argument, we get

$$B_\psi(\boldsymbol{x}_{t+1}, \boldsymbol{x}_t) \leq \langle \nabla\psi(\boldsymbol{x}_{t+1}) - \nabla\psi(\boldsymbol{x}_t), \boldsymbol{x}_{t+1} - \boldsymbol{x}_t \rangle \leq \langle \eta_t \boldsymbol{g}_t', \boldsymbol{x}_t - \boldsymbol{x}_{t+1} \rangle \leq \eta_t \|\boldsymbol{g}_t'\|_\star \|\boldsymbol{x}_{t+1} - \boldsymbol{x}_t\|$$

$$\leq \eta_t \|\boldsymbol{g}_t'\|_\star \sqrt{2 B_\psi(\boldsymbol{x}_{t+1}, \boldsymbol{x}_t)},$$

where we used Eq. (6) in the second inequality and Eq. (2) in the last one. Solving this inequality with respect to $B_\psi(\boldsymbol{x}_{t+1}, \boldsymbol{x}_t)$, we get $\sqrt{2B_\psi(\boldsymbol{x}_{t+1}, \boldsymbol{x}_t)} \leq 2\eta_t \|\boldsymbol{g}_t'\|_\star$.

For the second term, it suffices to subtract $B_\psi(\boldsymbol{x}_{t+1}, \boldsymbol{x}_t)/\eta_t$ on both sides of Eq. (12) and use the fact that $bx - \frac{a}{2}x^2 \leq \frac{b^2}{2a}, \forall x \in \mathbb{R}$ with $x = B_\psi(\boldsymbol{x}_{t+1}, \boldsymbol{x}_t)$. $\qquad\square$

This Theorem immediately gives us that Algorithm 1 has a regret upper-bounded by

$$R_T(\boldsymbol{u}) \leq \sum_{t=1}^{T} \frac{B_\psi(\boldsymbol{u}, \boldsymbol{x}_t) - B_\psi(\boldsymbol{u}, \boldsymbol{x}_{t+1})}{\eta_t} + \sum_{t=1}^{T} \eta_t \|\boldsymbol{g}_t\|_\star \min\left(2\|\boldsymbol{g}_t'\|_\star, \frac{\|\boldsymbol{g}_t\|_\star}{2}\right), \quad (13)$$

where $\boldsymbol{g}_t \in \partial \ell_t(\boldsymbol{x}_t)$. The presence of the minimum makes this bound equivalent in a worst-case sense to the one of OMD in Eq. (3). Moreover, at least in the Euclidean case, from Eq. (7) we have that $\|\boldsymbol{g}_t'\|_2 \leq \|\boldsymbol{g}_t\|_2$. However, it is difficult to quantify the gain over OMD because in general $\|\boldsymbol{g}_t\|_\star$ and $\|\boldsymbol{g}_t'\|_\star$ are data-dependent. Hence, as in the other previous analyses, the gain over OMD would be only marginal and not quantifiable. This is not a limit of our analysis: it is easy to realize that in the worst case the OMD update and the IOMD update can coincide. To show instead that a real gain is possible, we are now going to take a different path.

**Second Regret: Temporal Variability in IOMD.** Here we formalize our key intuition that IOMD is using an approximation of the future subgradient when the losses do not vary much over time. We use the notion of *temporal variability of the losses*, $V_T$, as given in Eq. (1). Considering again our regret bound in Theorem 5.2 and using $\eta_t = \eta$ for all $t$, we immediately have

$$R_T(\boldsymbol{u}) \leq \frac{B_\psi(\boldsymbol{u}, \boldsymbol{x}_1)}{\eta} + \sum_{t=1}^{T} \left(\ell_t(\boldsymbol{x}_t) - \ell_t(\boldsymbol{x}_{t+1}) - \frac{B_\psi(\boldsymbol{x}_{t+1}, \boldsymbol{x}_t)}{\eta}\right)$$

$$\leq \frac{B_\psi(\boldsymbol{u}, \boldsymbol{x}_1)}{\eta} + \ell_1(\boldsymbol{x}_1) - \ell_T(\boldsymbol{x}_{T+1}) + \sum_{t=2}^{T} \left(\max_{\boldsymbol{x} \in V} \ell_t(\boldsymbol{x}) - \ell_{t-1}(\boldsymbol{x}) - \frac{B_\psi(\boldsymbol{x}_{t+1}, \boldsymbol{x}_t)}{\eta}\right)$$

$$\leq \frac{B_\psi(\boldsymbol{u}, \boldsymbol{x}_1)}{\eta} + \ell_1(\boldsymbol{x}_1) - \ell_T(\boldsymbol{x}_{T+1}) + V_T .$$

This means that using a *constant learning rate yields a regret bound of $\mathcal{O}(V_T + 1)$, which might be better than $\mathcal{O}(\sqrt{T})$ if the temporal variability is low*. In particular, we can even get constant regret if $V_T = \mathcal{O}(1)$. On the contrary, OMD cannot achieve a constant regret for any convex loss even if $V_T = 0$, since it would imply an impossible $\mathcal{O}(1/T)$ rate for non-smooth batch black-box optimization [23, Theorem 3.2.1]. Instead, IOMD does not violate the lower bound since it is not a black-box method. As far as we know, the connection between IOMD and temporal variability has never been observed before. On the other hand, even when the temporal variability is high, we can still use a $\mathcal{O}(1/\sqrt{T})$ learning rate to achieve a worst case regret of the order $\mathcal{O}(\sqrt{T})$.

We would like to point out that a similar behaviour arises from *Follow The Regularized Leader* algorithm (FTRL) employed with full losses, rather than linearized ones. We show a detailed derivation in Appendix E. Unfortunately, contrarily to the OMD case employing FTRL would entail solving a constrained convex optimization problem whose size (in terms of number of functions) grows each step, that would have a high running time even when the implicit updates have closed form expressions, e.g., linear classification with hinge loss.

Finally, a natural question arises: can we get a bound which interpolates between $\mathcal{O}(V_T + 1)$ and $\mathcal{O}(\sqrt{T})$, without any prior knowledge on the quantity $V_T$? We give a positive answer to this question by presenting an adaptive strategy in the next section.

## 6 Adapting to the Temporal variability with AdaImplicit

In this section, we present an adaptive strategy to set the learning rates, in order to give a regret guarantee that depends optimally on the temporal variability.

From the previous section, we saw that the key quantity in the IOMD regret bound is

$$\delta_t \triangleq \ell_t(\boldsymbol{x}_t) - \ell_t(\boldsymbol{x}_{t+1}) - \frac{B_\psi(\boldsymbol{x}_{t+1}, \boldsymbol{x}_t)}{\eta_t} . \quad (14)$$

**Algorithm 2** AdaImplicit

---

**Require:** Non-empty closed convex set $V \subset X \subset \mathbb{R}^d$, $\psi : X \to \mathbb{R}$, $\lambda_1 = 0$, $\beta^2 > 0$, $\boldsymbol{x}_1 \in V$
1: **for** $t = 1, \ldots, T$ **do**
2:     Output $\boldsymbol{x}_t \in V$
3:     Receive $\ell_t : \mathbb{R}^d \to \mathbb{R}$ and pay $\ell_t(\boldsymbol{x}_t)$
4:     Update $\boldsymbol{x}_{t+1} = \arg\min_{\boldsymbol{x} \in V} \ \ell_t(\boldsymbol{x}) + \lambda_t B_\psi(\boldsymbol{x}, \boldsymbol{x}_t)$
5:     Set $\delta_t = \ell_t(\boldsymbol{x}_t) - \ell_t(\boldsymbol{x}_{t+1}) - \lambda_t B_\psi(\boldsymbol{x}_{t+1}, \boldsymbol{x}_t)$
6:     Update $\lambda_{t+1} = \lambda_t + \frac{1}{\beta^2} \delta_t$
7: **end for**

---

From Eq. (5), we have that $\delta_t \geq 0$. At this point, one might think of using a doubling trick: monitor $\sum_{i=1}^{t} \delta_i$ over time and restart the algorithm with a different learning rate once it exceeds a certain threshold. In Appendix A, we show that it is indeed possible to use such a strategy. However, while theoretically effective, we can't expect the doubling trick to have any decent performance in practice. Consequently, we are going to show how to use instead an *adaptive* learning rate.

**AdaImplicit.** Define $D^2 \triangleq \max_{\boldsymbol{x}, \boldsymbol{u} \in V} B_\psi(\boldsymbol{u}, \boldsymbol{x})$ and assume $D < \infty$. For ease of notation, we let $\eta_t = 1/\lambda_t$ where $\lambda_t$ will be decided in the following. Assuming $(\lambda_t)_{t=1}^T$ to be an increasing sequence, from Theorem 5.2 we get

$$R_T(\boldsymbol{u}) \leq D^2 \lambda_T + \sum_{t=1}^{T} \left[ \ell_t(\boldsymbol{x}_t) - \ell_t(\boldsymbol{x}_{t+1}) - \lambda_t B_\psi(\boldsymbol{x}_{t+1}, \boldsymbol{x}_t) \right] \ . \tag{15}$$

Ideally, to minimize the regret we would like to have $\lambda_T$ to be as close as possible to the sum over time in the r.h.s. of this expression. However, setting $\lambda_t \propto \sum_{s=1}^{t} \delta_i$ would introduce an annoying recurrence in the computation of $\lambda_t$. To solve this issue, we explore the same strategy adopted in AdaFTRL [25], adapting it to the OMD case: we set $\lambda_{t+1} = \frac{1}{\beta^2} \sum_{i=1}^{t} \delta_i$ for $t \geq 2$, for a parameter $\beta$ to be defined later, and $\lambda_1 = 0$. We call the resulting algorithm AdaImplicit and describe it in Algorithm 2. Before proving a regret bound for it, we first provide a technical lemma for the analysis. This lemma can be found in [24, 26] and for completeness we give a proof in Appendix B.

**Lemma 6.1.** *Let $\{a_t\}_{t=1}^\infty$ be any sequence of non-negative real numbers. Suppose that $\{\Delta_t\}_{t=1}^\infty$ is a sequence of non-negative real numbers satisfying $\Delta_1 = 0$ and[3] $\Delta_{t+1} \leq \Delta_t + \min\left\{ b a_t, \ c a_t^2/(2\Delta_t) \right\}$, for any $t \geq 1$. Then, for any $T \geq 0$, $\Delta_{T+1} \leq \sqrt{(b^2 + c) \sum_{t=1}^T a_t^2}$.*

We are now ready to prove a regret bound for Algorithm 2.

**Theorem 6.2.** *Let $V \subset X \subset \mathbb{R}^d$ be a non-empty closed convex set. Let $B_\psi$ be the Bregman divergence w.r.t. $\psi : X \to \mathbb{R}$ and let $D^2 = \max_{\boldsymbol{x}, \boldsymbol{u} \in V} B_\psi(\boldsymbol{u}, \boldsymbol{x})$. Assume $\psi$ to be 1-strongly convex with respect to $\|\cdot\|$ in $V$. Then, for any $\boldsymbol{u} \in V$, running Algorithm 2 with $\beta = D$ guarantees*

$$R_T(\boldsymbol{u}) \leq \min \left\{ 2(\ell_1(\boldsymbol{x}_1) - \ell_T(\boldsymbol{x}_{T+1}) + V_T), \ 2D\sqrt{3 \sum_{t=1}^T \|\boldsymbol{g}_t\|_\star^2} \right\}, \quad \forall \boldsymbol{g}_t \in \ell_t(\boldsymbol{x}_t) \ . \tag{16}$$

*Proof.* Using the definition of $\lambda_t$ and the fact that the sequence $(\lambda_t)_{t=1}^{T+1}$ is increasing over time, the regret in Eq. (15) can be upper bounded as $R_T(\boldsymbol{u}) \leq (D^2 + \beta^2) \lambda_{T+1}$. Therefore, we need an upper bound on $\lambda_{T+1}$. We split the proof in two parts, one for each term in the $\min$ in Eq. (16). For the first term, using the definition of $\lambda_t$ we have

$$\beta^2 \lambda_{T+1} = \sum_{t=1}^T [\ell_t(\boldsymbol{x}_t) - \ell_t(\boldsymbol{x}_{t+1}) - \lambda_t B_\psi(\boldsymbol{x}_{t+1}, \boldsymbol{x}_t)]$$

$$\leq \ell_1(\boldsymbol{x}_1) - \ell_T(\boldsymbol{x}_{T+1}) + \sum_{t=2}^T [\ell_t(\boldsymbol{x}_t) - \ell_{t-1}(\boldsymbol{x}_t)] \leq \ell_1(\boldsymbol{x}_1) - \ell_T(\boldsymbol{x}_{T+1}) + V_T \ ,$$

from which using $\beta = D$ the result follows.

For the second term, from Lemma 5.3 for $t \geq 2$ we have $\delta_t \leq \frac{\|\boldsymbol{g}_t\|_\star^2}{2\lambda_t}$. On the other hand,

$$\delta_t = \ell_t(\boldsymbol{x}_t) - \ell_t(\boldsymbol{x}_{t+1}) - \lambda_t B_\psi(\boldsymbol{x}_{t+1}, \boldsymbol{x}_t) \leq \ell_t(\boldsymbol{x}_t) - \ell_t(\boldsymbol{x}_{t+1}) \leq \langle \boldsymbol{g}_t, \boldsymbol{x}_t - \boldsymbol{x}_{t+1} \rangle$$

$$\leq \|\boldsymbol{g}_t\|_\star \|\boldsymbol{x}_t - \boldsymbol{x}_{t+1}\| \leq \sqrt{2}D\|\boldsymbol{g}_t\|_\star,$$

where in the last step we used Eq. (2) and the definition of $D$. Therefore, putting the last two results together we get

$$\delta_t \leq \min\left(\sqrt{2}D\|\boldsymbol{g}_t\|_\star, \|\boldsymbol{g}_t\|_\star^2/(2\lambda_t)\right), \quad \forall \boldsymbol{g}_t \in \partial \ell_t(\boldsymbol{x}_t).$$

Note that $\lambda_{t+1} = \lambda_t + \frac{1}{\beta^2}\delta_t$. Hence, $\lambda_1 = 0$, $\lambda_2 = (\ell_1(\boldsymbol{x}_1) - \ell_1(\boldsymbol{x}_2))/\beta^2 \leq \sqrt{2}D\|\boldsymbol{g}_1\|_\star/\beta^2$, and

$$\lambda_{t+1} = \lambda_t + \frac{1}{\beta^2}\delta_t \leq \lambda_t + \frac{1}{\beta^2}\min\left(\sqrt{2}D\|\boldsymbol{g}_t\|_\star, \frac{\|\boldsymbol{g}_t\|_\star^2}{2\lambda_t}\right), \quad \forall t \geq 3.$$

Therefore, using Lemma 6.1 with $\Delta_t = \lambda_t$, $b = \frac{\sqrt{2}D}{\beta^2}$ and $c = \frac{1}{\beta^2}$, $a_t = \|\boldsymbol{g}_t\|_\star$, we get

$$\lambda_{T+1} \leq \sqrt{(2D^2/\beta^4 + 1/\beta^2)\sum_{t=1}^{T}\|\boldsymbol{g}_t\|_\star^2},$$

from which setting $\beta = D$ we obtain the second term in the $\min$ in Eq. (16). $\square$

This last theorem shows that Algorithm 2 can have a low regret if the temporal variability of the losses $V_T$ is low. Moreover, differently from Optimistic Algorithms, Algorithm 2 does not need additional assumptions on the losses (for example smoothness), as done for example in [15].

**Lower Bound.** Next, we are going to prove a lower bound in terms of the temporal variability $V_T$, which shows that the regret bound in Theorem 6.2 cannot be improved further. The proof is a simple modification of the standard arguments used to prove lower bounds for constrained OLO and is reported in Appendix B.

**Theorem 6.3.** *Let $d \geq 2$, $\|\cdot\|$ an arbitrary norm on $\mathbb{R}^d$, and $V = \{\boldsymbol{x} \in \mathbb{R}^d : \|\boldsymbol{x}\| \leq D/2\}$. Let $\mathcal{A}$ be a deterministic algorithm on $V$. Let $T$ be any non-negative integer. Then, for any $V'_T \geq 0$, there exists a sequence of convex loss functions $\ell_1(\boldsymbol{x}), \ldots, \ell_T(\boldsymbol{x})$ with temporal variability equal to $V'_T$ and $\boldsymbol{u} \in V$ such that the regret of algorithm $\mathcal{A}$ satisfies $R_T(\boldsymbol{u}) \geq V'_T$.*

## 7 Empirical results

In this section, we compare the empirical performance of our algorithm AdaImplicit with standard baselines in online learning: OGD [38], OGD with adaptive learning rate $\eta_t = \frac{\beta}{\sqrt{\sum_{i=1}^{t}\|\boldsymbol{g}_i\|_\star^2}}$ (AdaOGD) [21], and IOMD with $\eta_t = \beta/\sqrt{t}$ (Implicit) [18].

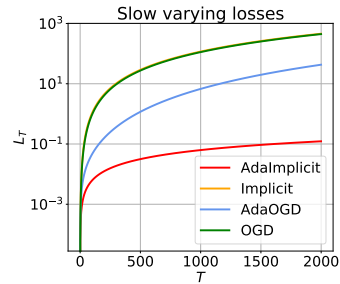

Figure 1: Synthetic experiment.

**Synthetic Experiment.** We first show the benefits of AdaImplicit on a synthetic dataset. The loss functions are chosen to have a small temporal variability $V_T$. In particular, we consider a 1-$d$ case using $\ell_t(x) = \frac{1}{4}(x - y_t)^2$ with $y_t = 100\sin(\pi\frac{t}{10T})$, a time horizon $T = 2000$ and the $L_2$ ball of diameter $D = 150$. We set $\beta = 1$ in all algorithms. The update of the implicit algorithms can be computed in closed form: $x_{t+1} = x_t - \frac{\eta_t}{2+\eta_t}(x_t - y_t)$. In Fig. 1 we show the cumulative loss $L_T = \sum_{t=1}^{T}\ell_t(x_t)$ of the algorithms (note that the $y$-axis is plotted in logarithmic scale). From the figure we can see that, contrarily to the other algorithms, the cumulative loss of AdaImplicit grows slowly over time, reflecting experimentally the bound given in Theorem 6.2. Also, even if not directly observable, OGD and IOMD basically incur the same total cumulative loss.

**Real world datasets.** We are now going to show some experiments conducted on real data. Here, there is no reason to believe that the temporal variability is small. However, we still want to verify

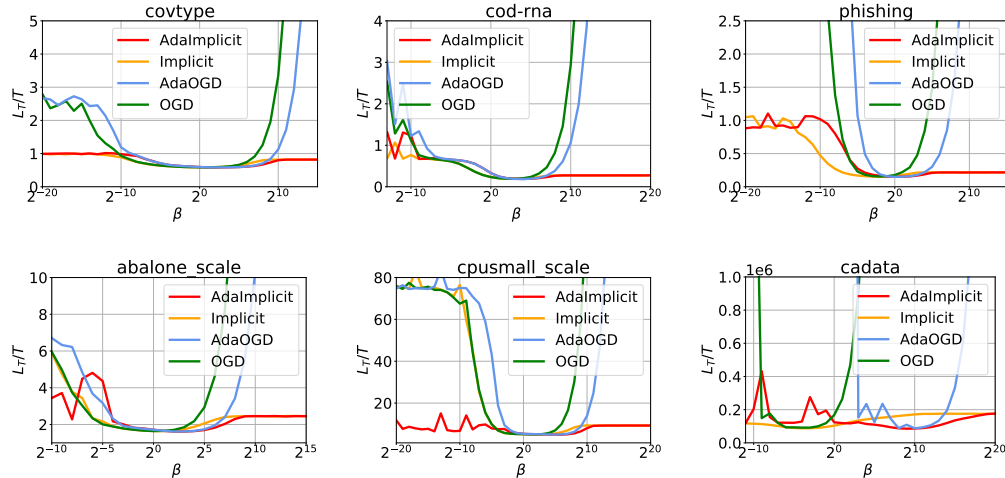

Figure 2: Plots on classification tasks using the hinge loss (top) and regression tasks using the absolute loss (bottom).

if AdaImplicit can achieve a good worst-case performance. We consider both classification and regression tasks. Additional plots can be found in Appendix D.

We used datasets from the LIBSVM library [6]. Before running the algorithms, we preprocess the data by dividing each feature by its maximum absolute value so that all the values are in the range $[-1, 1]$, then we add a bias term. Details about the datasets can be found in Appendix D.

Given that in the online setting we cannot tune the hyperparameter $\beta$ using hold-out data, we plot the average cumulative loss of each algorithm, i.e., $L_t/t = \frac{1}{t}\sum_{i=1}^{t}\ell_i(\boldsymbol{x}_i)$, as a function of the hyperparameter $\beta$. This allows us to evaluate at the same time the sensitivity of the algorithms to $\beta$ and their best performance with oracle tuning. Note that in all the algorithms we consider the optimal worst-case setting of $\beta$ is proportional to the diameter of the feasible set, hence it is fair to plot their performance as a function of $\beta$. We consider values of $\beta$ in $[2^{-20}, 2^{20}]$ with a grid containing 41 points. Then, each algorithm is run 10 times and results are averaged. For classification tasks we use the hinge loss, while for regression tasks we use the absolute loss. In both cases, we adopt the squared $L_2$ function for $\psi$. The details about implicit updates are discussed in Appendix C.

Results are illustrated in Fig. 2. From the plots, we can see that when fine-tuned, all the algorithms achieve similar results, i.e., the minimum value of average cumulative loss is very close for all the algorithms considered and there is not a clear winner. However, note that the range of values which allows an algorithm to reach the minimum is considerably wider for Implicit algorithms and confirms their robustness regarding learning rate misspecification, as already investigated in other works [see, e.g., 33, 34]. This is a great advantage when considering online algorithms since, contrarily to the batch setting, algorithms cannot be fine-tuned in advance relying on training/validation sets.

# 8 Conclusions

In this paper, we investigated online Implicit algorithms from a theoretical perspective. Our analysis revealed interesting insights regarding the behavior of these algorithms and allowed us to design a new *adaptive* algorithm, which may take advantage of "easy" data. The obtained experimental results indicate that in real-world tasks (such as online classification with hinge loss or online regression with the absolute loss), Implicit algorithms provide a better solution in terms of robustness, which is particularly relevant in online settings. Future directions include extending our analysis to a broader area, for example considering dynamic environments or strongly-convex loss functions, to see if the same gains can be proved. Finally, other examples of "easy" data can be considered, such as the case of stochastic loss functions.

## Broader Impact

We believe our investigation will foster further studies promoting the adoption of adaptive learning rates in online learning and beyond. Indeed, in recent years adaptive methods in optimization proved to be one of the preferred methods for training deep neural networks. On the other hand, this work confirm the robustness of implicit updates and opens up to new possibilities in this field. From a societal aspect, this work in mainly theoretical and does not present any foreseeable consequence.

## Acknowledgements

This material is based upon work supported by the National Science Foundation under grants no. 1925930 "Collaborative Research: TRIPODS Institute for Optimization and Learning" and no. 1908111 "AF: Small: Collaborative Research: New Representations for Learning Algorithms and Secure Computation". NC thanks Nicolò Cesa-Bianchi for supporting his visit to Boston University.

## Footnotes

[2]Eq. (7) is nothing else than the fact that subgradients are monotone operators.

[3]With a small abuse of notation, let $\min(x, y/0) = x$.

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
