[Supplementary Material]

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

    Lipschitz constant $L > 0$, $\beta > 0$, $\boldsymbol{x}_1 \in V$
1:  Initialize: $i = 0$, $S_0 = 0$, $\eta_0 = \frac{\beta}{L}$
2:  **for** $t = 1, \ldots, T$ **do**
3:     Output $\boldsymbol{x}_t$
4:     Receive $\ell_t : \mathbb{R}^d \to (-\infty, +\infty]$ and pay $\ell_t(\boldsymbol{x}_t)$
5:     $\boldsymbol{x}'_{t+1} = \arg\min_{\boldsymbol{x} \in V} \ B_\psi(\boldsymbol{x}, \boldsymbol{x}_t) + \eta_i \ell_t(\boldsymbol{x})$
6:     $\delta_t = \ell_t(\boldsymbol{x}_t) - \ell_t(\boldsymbol{x}'_{t+1}) - \frac{B_\psi(\boldsymbol{x}'_{t+1} - \boldsymbol{x}_t)}{\eta_i}$
7:     $S_i \leftarrow S_i + \delta_t$
8:     **if** $S_i \geq \eta_i L^2 2^i$ **then**
9:        $i = i + 1$
10:      Update $\eta_i = \frac{\beta}{L\sqrt{2^i}}$
11:       $S_i = 0$
12:       $\boldsymbol{x}_{t+1} = \boldsymbol{x}_1$
13:     **else**
14:       $\boldsymbol{x}_{t+1} = \boldsymbol{x}'_{t+1}$
15:     **end if**
16: **end for**

---

## A   Doubling Trick

In this section, we present a doubling trick strategy to tune the learning rate in IOMD. As already mentioned in Section 6, it is possible to apply this construction if we consider the terms $\delta_t$, whose sum over time is an increasing sequence thanks to Eq. (5). We present the doubling trick here to show that it is not simpler than our main analysis in the paper nor it adds empirical advantages. The only advantage coming from using a doubling trick derives from the fact that it is not required to have a bounded domain (as we will show in Theorem A.2), opposed to the case of an adaptive learning rate.

In the following, we slightly modify the definition of $\delta_t$ given in Eq. (14) (see line 6 of Algorithm 3),

$$\delta_t = \ell_t(\boldsymbol{x}_t) - \ell_t(\boldsymbol{x}'_{t+1}) - \frac{B_\psi(\boldsymbol{x}_t, \boldsymbol{x}'_{t+1})}{\eta_t} \ .$$

The algorithm works as follows: at the beginning of epoch $i$, we set the learning rate $\eta_i = \beta/L\sqrt{2^i}$, run IOMD and monitor the sum $\sum_t \delta_t/\eta_i$ until it reaches $L^2 2^i$. Once this happens, we restart the algorithm doubling the threshold and halving the learning rate—see Algorithm 3. Note that during an epoch the learning rate stays fixed. Let $t_i$ be the index of the first round of epoch $i$. We then have $\eta_t = \eta_i$ for $t \in [t_i, t_{i+1} - 1]$.

For the analysis, note that by using a doubling trick the time horizon is divided in $N$ different epochs (where $N$ is obviously not known a priori). We next show that the total number of epochs $N$ is logarithmic in a quantity $\Delta_i$ which we define for the sake of the analysis.

**Lemma A.1.** *Let $t_i$ be the first time-step of epoch $i$, with $t_0 = 1$. Suppose Algorithm 3 is run for a total of $N$ epochs. Let $\Delta_i \triangleq \sum_{t=t_i}^{t_{i+1}-1} \delta_t$. Then, we have that*

$$N \leq \min \left( \log_2 \left( \frac{1}{L^2} \sum_{i=0}^{N} \frac{\Delta_i}{\eta_i} + 1 \right), 2 \log_2 \left( \frac{\sqrt{2} - 1}{L} \sum_{i=0}^{N} \Delta_i + 1 \right) \right) \ . \tag{17}$$

*Proof.* First, we have that

$$\sum_{i=0}^{N-1} a^i = \frac{a^N - 1}{a - 1} \ . \tag{18}$$

$$\leq \frac{\Delta_2}{\eta_2} \leq L^2 \left(2^2 + \frac{1}{2}\right)$$

Figure 3: Doubling trick illustrated.

Now, note that the sum of the $L^2 2^i$ terms in the first $N-1$ epochs is less than or equal to the final sum of the terms monitored. Therefore, using Eq. (18) we have

$$\sum_{i=0}^{N-1} L^2 2^i = L^2(2^N - 1) \leq \sum_{i=0}^{N} \sum_{t=t_i}^{t_{i+1}-1} \frac{\delta_t}{\eta_i} = \sum_{i=0}^{N} \frac{\Delta_i}{\eta_i},$$

and solving for $N$ yields the first term in the min in Eq. (17).

On the other hand, we have that the condition for the restart can be rewritten as $\sum_t \delta_t \leq L\sqrt{2^i}$. Using again Eq. (18), we get

$$\sum_{i=0}^{N-1} L\sqrt{2^i} = L\frac{2^{\frac{N}{2}} - 1}{\sqrt{2} - 1} \leq \sum_{i=0}^{N} \sum_{t=t_i}^{t_{i+1}-1} \delta_t = \sum_{i=0}^{N} \Delta_i, \tag{19}$$

rearranging and solving for $N$ gives the second term in the min in Eq. (17). $\square$

We can now analyze the regret incurred by the algorithm. Each epoch is bounded individually. The final regret bound is given by the sum of the individual contributions over all the epochs.

**Theorem A.2.** *Assume the losses $\ell_t(\boldsymbol{x})$ to be $L$-Lipschitz, for all $t = 1, \ldots, T$ and $\psi$ to be 1-strongly convex w.r.t. $\|\cdot\|$. Let $\Delta_i$ be defined as in Lemma A.1. Then, for any $\boldsymbol{u} \in \mathbb{R}^d$ the regret of Algorithm 3 after $T$ rounds is bounded as*

$$R_T(\boldsymbol{u}) \leq c\left(\frac{B_\psi(\boldsymbol{u}, \boldsymbol{x}_1)}{\beta} + \beta\right) L\sqrt{T+1} + c\frac{\beta L}{2}, \tag{20}$$

*where $c = \frac{\sqrt{2}}{\sqrt{2}-1}$.*

*Proof.* Using the notation introduced, we have that $\delta_{t_{i+1}-1}$ is the term which causes the restart in epoch $i$. Therefore, the following holds

$$\frac{\Delta_i}{\eta_i} \leq L^2 \left(2^i + \frac{1}{2}\right), \tag{21}$$

since from Lemma 5.3 the last term in epoch $i$ is such that $\frac{\delta_{t_{i+1}-1}}{\eta_i} \leq \frac{L^2}{2}$.

We next define $R_i(\boldsymbol{u})$ as the regret during epoch $i$,

$$R_i(\boldsymbol{u}) = \sum_{t=t_i}^{t_{i+1}-1} (\ell_t(\boldsymbol{x}_t) - \ell_t(\boldsymbol{u})).$$

Using Eq. (10) with $\eta_t = \eta_i$ for $t \in [t_i, \ldots, t_{i+1} - 1]$, we have that $R_i(\boldsymbol{u})$ is bounded as follows

$$R_i(\boldsymbol{u}) \leq \frac{B_\psi(\boldsymbol{u}, \boldsymbol{x}_{t_i})}{\eta_i} + \sum_{t=t_i}^{t_{i+1}-1} \left[\ell_t(\boldsymbol{x}_t) - \ell_t(\boldsymbol{x}'_{t+1}) - \frac{1}{\eta_i} B_\psi(\boldsymbol{x}'_{t+1}, \boldsymbol{x}_t)\right]$$

$$\leq \frac{B_\psi(\boldsymbol{u}, \boldsymbol{x}_{t_i})}{\eta_i} + \sum_{t=t_i}^{t_{i+1}-1} \delta_t = \frac{B_\psi(\boldsymbol{u}, \boldsymbol{x}_1)}{\eta_i} + \Delta_i . \tag{22}$$

We can now write the final regret bound by summing $R_i(\boldsymbol{u})$ over all the epochs $i = 1, \ldots, N$.

$$
\begin{aligned}
R_T(\boldsymbol{u}) &= \sum_{i=0}^{N} R_i(\boldsymbol{u}) \\
&\leq \sum_{i=0}^{N} \left[ \frac{B_\psi(\boldsymbol{u}, \boldsymbol{x}_1)}{\eta_i} + \Delta_i \right] \\
&\leq \sum_{i=0}^{N} \left[ \frac{B_\psi(\boldsymbol{u}, \boldsymbol{x}_1)}{\eta_i} + \eta_i L^2 \left( 2^i + \frac{1}{2} \right) \right] \qquad\qquad \text{see Eq. (21)} \\
&= L \left( \frac{B_\psi(\boldsymbol{u}, \boldsymbol{x}_1)}{\beta} + \beta \right) \sum_{i=0}^{N} \sqrt{2^i} + \frac{\beta L}{2} \sum_{i=0}^{N} \frac{1}{\sqrt{2^i}} \qquad \left( \eta_i = \frac{\beta}{L\sqrt{2^i}} \right) \\
&\leq \left( \frac{B_\psi(\boldsymbol{u}, \boldsymbol{x}_1)}{\beta} + \beta \right) L \frac{2^{\frac{N+1}{2}} - 1}{\sqrt{2} - 1} + \frac{\beta L}{2} \frac{\sqrt{2}}{\sqrt{2} - 1} \\
&\leq c \left( \frac{B_\psi(\boldsymbol{u}, \boldsymbol{x}_1)}{\beta} + \beta \right) L 2^{\frac{N}{2}} + c \frac{\beta L}{2},
\end{aligned}
\tag{23}
$$

where in the last step we used Eq. (18) and the definition of $c$.

Now, using the first term in Eq. (17) we have that

$$
2^{\frac{N}{2}} = 2^{\frac{1}{2} \log_2 \left( \frac{1}{L^2} \sum_{i=0}^{N} \frac{\Delta_i}{\eta_i} + 1 \right)} = \sqrt{\frac{1}{L^2} \sum_{i=0}^{N} \frac{\Delta_i}{\eta_i} + 1} \ .
$$

Therefore, from Eq. (23) the regret can be bounded as

$$
R_T(\boldsymbol{u}) \leq c \left( \frac{B_\psi(\boldsymbol{u}, \boldsymbol{x}_1)}{\beta} + \beta \right) \sqrt{\sum_{i=0}^{N} \frac{\Delta_i}{\eta_i} + L^2} + c \frac{\beta L}{2} \ .
$$

Furthermore, using Theorem 5.3 and the assumption on the losses to be $L$-Lipschitz, we get

$$
\sum_{i=0}^{N} \frac{\Delta_i}{\eta_i} = \sum_{i=0}^{N} \sum_{t=t_i}^{t_{i+1}-1} \frac{\delta_t}{\eta_i} \leq \sum_{i=0}^{N} \sum_{t=t_i}^{t_{i+1}-1} \frac{L^2}{2} = \frac{L^2}{2} T,
$$

substituting back the above result we get

$$
R_T(\boldsymbol{u}) \leq cL \left( \frac{B_\psi(\boldsymbol{u}, \boldsymbol{x}_1)}{\beta} + \beta \right) \sqrt{T+1} + c \frac{\beta L}{2} \ . \qquad\qquad \square
$$

In principle, it is possible to get a bound which interpolates between $\mathcal{O}(\sqrt{T})$ and $\mathcal{O}(V_T)$ as done in Theorem 6.2. However, for the latter possibility a bounded domain seems required, in order to bound the difference between the first and last losses of each epoch. Moreover, the number of restarts is in the worst case order of $\mathcal{O}(\ln T)$, which would give a (slightly) worse bound compared to Theorem 6.2. For these reasons, we will not provide details about how to use Algorithm 3 in order to get a regret bound order of $\mathcal{O}(V_T)$. Nonetheless, we next show how in the case of fixed losses, we can still recover a constant regret bound, even if the domain is unbounded.

## A.1 Fixed Losses

If the losses are all equal, i.e. $\ell_t(\boldsymbol{x}) = \ell(\boldsymbol{x})$ for all $t = 1, \ldots, T$, running Algorithm 3 from Theorem A.2 we would expect a regret which scales as $\mathcal{O}(\sqrt{T})$. However, as we are going to show in the next lemma, with a proper setting of $\beta$ this is actually not the case and Algorithm 3 always stays in the first epoch, even if the domain is unbounded!

**Lemma A.3.** *Assume the losses over time are fixed, i.e. $\ell_t(\boldsymbol{x}) = \ell(\boldsymbol{x})$ for all $t = 1, \ldots, T$ and $\psi(\boldsymbol{x})$ be 1-strongly convex w.r.t $\| \cdot \|$. Then, Algorithm 3 with $\beta \geq 1$ will always stay in the first epoch, i.e., $N = 0$.*

*Proof.* In order to not have a restart, we need $\delta_t \leq \eta_i L^2 2^i$, which translates to

$$\ell(\boldsymbol{x}_t) - \ell(\boldsymbol{x}'_{t+1}) \leq \beta L \sqrt{2^i} \left( B_\psi(\boldsymbol{x}'_{t+1}, \boldsymbol{x}_t) + 1 \right) .$$

From the inequality above, we have a reset iff

$$\begin{aligned} i &\leq 2 \log_2 \frac{\ell(\boldsymbol{x}_t) - \ell(\boldsymbol{x}'_{t+1})}{\beta L \left( B_\psi(\boldsymbol{x}'_{t+1}, \boldsymbol{x}_t) + 1 \right)} \leq 2 \log_2 \frac{L \|\boldsymbol{x}_t - \boldsymbol{x}'_{t+1}\|}{\beta L \left( 1 + \frac{1}{2} \|\boldsymbol{x}'_{t+1} - \boldsymbol{x}_t\|^2 \right)} \\ &= 2 \log_2 \frac{2\|\boldsymbol{x}'_{t+1} - \boldsymbol{x}_t\|}{\beta(2 + \|\boldsymbol{x}'_{t+1} - \boldsymbol{x}_t\|^2)}, \end{aligned}$$

where the second inequality derives from the Lipschitzness of the losses and by lower bounding the Bregman divergence with Eq. (2). Now, let $f(y) = \frac{2y}{2+y^2}$, with $y \geq 0$. We have that $\lim_{y \to +\infty} f(y) = 0$ and $f(0) = 0$. Furthermore, if we take the derivative and set it to 0, we see that $f(y)$ has a maximum in $y = \sqrt{2}$. Hence, we have

$$i \leq 2 \log_2 \frac{2\|\boldsymbol{x}'_{t+1} - \boldsymbol{x}_t\|}{\beta(2 + \|\boldsymbol{x}'_{t+1} - \boldsymbol{x}_t\|^2)} \leq 2 \log_2 \frac{1}{\beta\sqrt{2}} = -1 - 2\log_2 \beta \leq -1,$$

where the last step derives from the assumption on $\beta$. Therefore, if $i > -1$ then we will not double the learning rate. Note that this is always verified and we can conclude that $N = 0$. $\square$

We can now prove a regret bound in the case of fixed losses.

**Theorem A.4.** *Under the assumptions of lemma A.3 the regret incurred by Algorithm 3 is bounded as*

$$R_T(\boldsymbol{u}) \leq \frac{L}{\beta} B_\psi(\boldsymbol{u}, \boldsymbol{x}_1) + \ell(\boldsymbol{x}_1) - \ell(\boldsymbol{x}_T) . \tag{24}$$

*Proof.* From Eq. (22) and the fact that we only have one epoch thanks to lemma A.3, we have that

$$\begin{aligned} R_T(\boldsymbol{u}) = R_0(\boldsymbol{u}) &\leq \frac{B_\psi(\boldsymbol{u}, \boldsymbol{x}_1)}{\eta_0} + \sum_{t=1}^{T} \delta_t \\ &= \frac{B_\psi(\boldsymbol{u}, \boldsymbol{x}_1)}{\eta_0} + \sum_{t=1}^{T} \left[ \ell(\boldsymbol{x}_t) - \ell(\boldsymbol{x}_{t+1}) - \frac{B_\psi(\boldsymbol{x}_{t+1}, \boldsymbol{x}_t)}{\eta_0} \right] \\ &\leq \frac{B_\psi(\boldsymbol{u}, \boldsymbol{x}_1)}{\eta_0} + \ell(\boldsymbol{x}_1) - \ell(\boldsymbol{x}_T) \\ &= \frac{L}{\beta} B_\psi(\boldsymbol{u}, \boldsymbol{x}_1) + \ell(\boldsymbol{x}_1) - \ell(\boldsymbol{x}_T) . \end{aligned}$$
$\square$

# B  Proofs

The proof of the properties in Proposition 4.1 is straightforward, but we report it here for completeness.

*Proof of Proposition 4.1.* From the update in Eq. (4), we immediately get the following inequality

$$\eta_t \ell_t(\boldsymbol{x}_{t+1}) \leq \eta_t \ell_t(\boldsymbol{x}_{t+1}) + B_\psi(\boldsymbol{x}_{t+1}, \boldsymbol{x}_t) \leq \eta_t \ell_t(\boldsymbol{x}_t) + B_\psi(\boldsymbol{x}_t, \boldsymbol{x}_t) = \eta_t \ell_t(\boldsymbol{x}_t),$$

which verifies Eq. (5) and implies that value of the loss $\ell_t$ in $\boldsymbol{x}_{t+1}$ is not bigger than the one in $\boldsymbol{x}_t$.

Eq. (6) is simply the first-order optimality condition for $\boldsymbol{x}_{t+1}$.

For Eq. (7), from the convexity of $\ell_t$ we have

$$\begin{aligned} \ell_t(\boldsymbol{x}_{t+1}) &\geq \ell_t(\boldsymbol{x}_t) + \langle \boldsymbol{g}_t, \boldsymbol{x}_{t+1} - \boldsymbol{x}_t \rangle, \\ \ell_t(\boldsymbol{x}_t) &\geq \ell_t(\boldsymbol{x}_{t+1}) + \langle \boldsymbol{g}'_t, \boldsymbol{x}_t - \boldsymbol{x}_{t+1} \rangle . \end{aligned}$$

Summing both inequalities, we get the desired result. $\square$

*Proof of Lemma 5.1.* From Eq. (6), we have that

$$\eta_t(\ell_t(\boldsymbol{x}_{t+1}) - \ell_t(\boldsymbol{u})) \leq \langle \eta_t \boldsymbol{g}'_t, \boldsymbol{x}_{t+1} - \boldsymbol{u} \rangle \leq \langle \nabla \psi(\boldsymbol{x}_t) - \nabla \psi(\boldsymbol{x}_{t+1}), \boldsymbol{x}_{t+1} - \boldsymbol{u} \rangle$$
$$= B_\psi(\boldsymbol{u}, \boldsymbol{x}_t) - B_\psi(\boldsymbol{u}, \boldsymbol{x}_{t+1}) - B_\psi(\boldsymbol{x}_{t+1}, \boldsymbol{x}_t) \ .$$

Dividing by $\eta_t$ and summing over $t = 1, \ldots, T$ yields the result in Eq. (8).

For the second part observe that

$$\sum_{t=1}^{T} \frac{B_\psi(\boldsymbol{u}, \boldsymbol{x}_t) - B_\psi(\boldsymbol{u}, \boldsymbol{x}_{t+1})}{\eta_t} \leq \frac{D^2}{\eta_1} + D^2 \sum_{t=2}^{T} \left( \frac{1}{\eta_t} - \frac{1}{\eta_{t-1}} \right) = \frac{D^2}{\eta_T} \ . \qquad \square$$

*Proof of Lemma 6.1.* From the assumptions, we have that $\Delta_{t+1} \leq \Delta_t + \min\{ba_t, \ ca_t^2/(2\Delta_t)\}$, for any $t \geq 1$. Also, observe that

$$\Delta_{T+1}^2 = \sum_{t=1}^{T} \Delta_{t+1}^2 - \Delta_t^2 = \sum_{t=1}^{T} \left[ \underbrace{(\Delta_{t+1} - \Delta_t)^2}_{(a)} + \underbrace{2(\Delta_{t+1} - \Delta_t)\Delta_t}_{(b)} \right] \ .$$

We bound (a) and (b) separately. For (a), from the assumption on the recurrence and using the first term in the minimum we have that $(\Delta_{t+1} - \Delta_t)^2 \leq b^2 a_t^2$. On the other hand, for (b) using the second term in the minimum in the recurrence we get $2(\Delta_{t+1} - \Delta_t)\Delta_t \leq ca_t^2$. Putting together the results we have that $\Delta_{T+1}^2 \leq (b^2 + c) \sum_{t=1}^{T} a_t^2$ and the lemma follows. $\qquad \square$

### B.1 Lower Bound

*Proof of Theorem 6.3.* The first loss of the algorithm is $\ell_1(\boldsymbol{x}) = L\langle \boldsymbol{g}, \boldsymbol{x} \rangle$, where $\|\boldsymbol{g}\|_\star = 1$ and $\boldsymbol{g}$ is orthogonal to $\boldsymbol{x}_1$, while $L$ will be set in the following. Note that $d \geq 2$ assures that $\boldsymbol{g}$ always exists. For $t \geq 2$, set $\ell_t(\boldsymbol{x}) = 0$. First, observe that $V_T = \max_{\boldsymbol{x} \in V} -\ell_1(\boldsymbol{x}) = L\max_{\boldsymbol{u} \in V} -\langle \boldsymbol{g}, \boldsymbol{u} \rangle = LD/2$. Hence, setting $L = 2V'_T/D$, we have $V_T = V'_T$. Also, we have $R_T(\boldsymbol{u}) = L\max_{\boldsymbol{u} \in V} -\langle \boldsymbol{g}, \boldsymbol{u} \rangle = V_T$. $\qquad \square$

It is worth emphasizing that the lower bound does not contradict the upper bound of $\mathcal{O}(DL\sqrt{T})$ because here $L$ is chosen arbitrarily large.

## C Formulas

First, let's mention the update rules for IOMD with $V = \mathbb{R}^d$ for hinge loss, absolute loss, and square loss respectively [see, e.g., 11, 18].

$$\boldsymbol{x}_{t+1} = \boldsymbol{x}_t + \min \left( \eta_t, \frac{\max(1 - y_t \langle \boldsymbol{z}_t, \boldsymbol{x}_t \rangle, 0)}{\|\boldsymbol{z}_t\|^2} \right) y_t \boldsymbol{z}_t,$$

$$\boldsymbol{x}_{t+1} = \boldsymbol{x}_t - \min \left( \eta_t, \frac{|\langle \boldsymbol{z}_t, \boldsymbol{x}_t \rangle - y_t|}{\|\boldsymbol{z}_t\|^2} \right) \boldsymbol{z}_t,$$

$$\boldsymbol{x}_{t+1} = \boldsymbol{x}_t - \eta_t \frac{(\langle \boldsymbol{z}_t, \boldsymbol{x}_t \rangle - y_t)\boldsymbol{z}_t}{1 + \eta \|\boldsymbol{z}_t\|_2^2} \ .$$

Now let's consider the case that $V = \{\boldsymbol{x} : \|\boldsymbol{x}\|_2 \leq D/2\}$. In this case it is easy to see that the update becomes

$$\boldsymbol{x}_{t+1} = \boldsymbol{x}_t - \eta_t \boldsymbol{g}'_t - \alpha \boldsymbol{x}_{t+1},$$

where $\boldsymbol{g}'_t \in \partial \ell_t(\boldsymbol{x}_{t+1})$, $\alpha \boldsymbol{x}_{t+1} \in \partial i_V(\boldsymbol{x}_{t+1})$ and $\alpha \geq 0$. Hence, we have that

$$\boldsymbol{x}_{t+1} = \frac{\boldsymbol{x}_t}{\alpha + 1} - \frac{\eta_t}{\alpha + 1} \boldsymbol{g}'_t \ .$$

This implies that we can take the previous formulas and substitute $\frac{\boldsymbol{x}_t}{\alpha+1}$ to $\boldsymbol{x}_t$ and $\frac{\eta_t}{\alpha+1}$ to $\eta_t$. Then, the optimal $\alpha \geq 0$ is the smallest one that gives $\|\boldsymbol{x}_{t+1}\| \leq D/2$. Note that this is a 1-dimensional problem that can be easily solved numerically.

Figure 4: Plots on classification tasks using the hinge loss (top) and regression tasks using the absolute loss (bottom).

## D Experiments

In Fig. 4 we show plots about other experiments on real data which were not shown in the main paper. Details about the datasets used can be found in Table 1 and Table 2.

Table 1: Classification datasets

| Name | Datapoints | Features |
|---|---|---|
| a9a | 32,561 | 123 |
| ijcnn1 | 49,990 | 22 |
| cod-rna | 59,535 | 8 |
| covtype | 581,012 | 54 |
| skin_nonskin | 245,057 | 3 |
| phishing | 11,055 | 68 |

Table 2: Regression datasets

| Name | Datapoints | Features |
|---|---|---|
| abalone | 11,055 | 8 |
| cadata | 20,640 | 8 |
| cpusmall | 8,192 | 12 |
| housing | 506 | 13 |
| space_ga | 3,107 | 6 |

## E Implicit Updates for FTRL

In this section, we show how to get a bound of $\mathcal{O}(V_T + 1)$ for FTRL employed with full losses. As already explained in the main paper, contrarily to the OMD case, we do not have efficient algorithms to solve the minimization problem given by the FTRL update rule in this case. Furthermore, we show that it is not possible to adopt the same learning rate tuning strategy of *AdaImplicit* in order to get a similar regret bound.

We first remember the FTRL regret bound, which is standard in the literature (see e.g. [24] Lemma 7.1).

**Theorem E.1.** *Let $V \subseteq \mathbb{R}^d$ be closed and non-empty. Denote by $F_t(\boldsymbol{x}) = \psi_t(\boldsymbol{x}) + \sum_{i=1}^{t-1} \ell_i(\boldsymbol{x})$, where $\psi_1, \ldots, \psi_T$ is a sequence of regularizers such that $\psi_t : \mathbb{R}^d \to (-\infty, +\infty]$ for all t. Assume that $\operatorname{argmin}_{\boldsymbol{x} \in V}$ is not empty and set $\boldsymbol{x}_1 = \operatorname{argmin}_{\boldsymbol{x} \in V} \psi_1(\boldsymbol{x})$, $\boldsymbol{x}_{t+1} \in \operatorname{argmin}_{\boldsymbol{x} \in V} F_t(\boldsymbol{x})$. Then, for any $\boldsymbol{u}$, we have*

$$R_T(\boldsymbol{u}) \leq (\psi_T(\boldsymbol{u}) - \psi_T(\boldsymbol{x}_1)) + \sum_{t=1}^{T} [F_t(\boldsymbol{x}_t) - F_{t+1}(\boldsymbol{x}_{t+1}) + \ell_t(\boldsymbol{x}_t)] . \qquad (25)$$

Now, assume that $\psi_t(\boldsymbol{x}) = \lambda_t \psi(\boldsymbol{x})$, with $(\lambda_t)_{t=1}^T$ being a non-decreasing sequence. We can rewrite the sum over time on the right-hand side of Eq. (25) as follows

$$\sum_{t=1}^T [F_t(\boldsymbol{x}_t) - F_{t+1}(\boldsymbol{x}_{t+1}) + \ell_t(\boldsymbol{x}_t)]$$

$$= \sum_{t=1}^T [F_t(\boldsymbol{x}_t) + \ell_t(\boldsymbol{x}_t) - F_t(\boldsymbol{x}_{t+1}) - \ell_t(\boldsymbol{x}_{t+1}) + \psi_t(\boldsymbol{x}_{t+1}) - \psi_{t+1}(\boldsymbol{x}_{t+1})]$$

$$\leq \sum_{t=1}^T [\ell_t(\boldsymbol{x}_t) - \ell_t(\boldsymbol{x}_{t+1}) + \psi_t(\boldsymbol{x}_{t+1}) - \psi_{t+1}(\boldsymbol{x}_{t+1})]$$

$$= \ell_1(\boldsymbol{x}_1) - \ell_T(\boldsymbol{x}_{T+1}) + \sum_{t=2}^T [\ell_t(\boldsymbol{x}_t) - \ell_{t-1}(\boldsymbol{x}_t)] + \sum_{t=1}^T (\lambda_t - \lambda_{t+1})\psi(\boldsymbol{x}_{t+1})$$

$$\leq \ell_1(\boldsymbol{x}_1) - \ell_T(\boldsymbol{x}_{T+1}) + V_T + \sum_{t=1}^T (\lambda_t - \lambda_{t+1})\psi(\boldsymbol{x}_{t+1}) \,,$$

where the first inequality derives from the fact that $F_t(\boldsymbol{x}_t) - F_t(\boldsymbol{x}_{t+1}) \leq 0$ while the last one from the definition of $V_T$. Therefore, the regret bound can be rewritten as follows

$$R_T(\boldsymbol{u}) \leq \lambda_T(\psi(\boldsymbol{u}) - \psi(\boldsymbol{x}_1)) + \ell_1(\boldsymbol{x}_1) - \ell_T(\boldsymbol{x}_{T+1}) + V_T + \sum_{t=1}^T (\lambda_t - \lambda_{t+1})\psi(\boldsymbol{x}_{t+1})$$

We can see that with a constant learning rate the above expression would give a regret bound $\mathcal{O}(V_T + 1)$. On the other hand, it is known from the literature that a parameter $\lambda_t \propto \sqrt{t}$ would give a regret bound of $\mathcal{O}(\sqrt{T})$. Ideally, we would like to have a certain $\lambda_t$ which allows to interpolate between a regret bound of $\mathcal{O}(V_T + 1)$ and $\mathcal{O}(\sqrt{T})$, as done for *AdaImplicit*. However, the techniques adopted in Section 6 do not seem to work in this case and one should hence resort to a different approach. In addition to the technical difficulties, as already stated in the main paper the computational burden of implicit updates with FTRL could be prohibitive in practice and makes this approach not worth of pursuing.