[Reviews · NeurIPS 2020]

Review 1

Summary and Contributions: This paper considers the implicit update algorithm for online learning. It is well known that this algorithm works very well in practice, although theoretical understanding of its benefits over online mirror descent are somewhat lacking. This paper shows that the implicit update algorithm enjoys a regret bound that is adapted to the variability of the sequence of loss functions. This regret bound holds even if the learning rate used is a constant. At the same time, the method has a worst case regret bound of O(sqrt{T}) by choosing a learning rate of O(1/sqrt{T}). To overcome this tuning of the learning rate, the authors also give an adaptive version of the implicit update method, called AdaImplicit, which uses techniques from the work of Orabona and Pal (2015) to careful change learning rates and automatically achieve the required regret bounds without knowledge of the variability of the loss functions. Crucially, this algorithm doesn't make use of the doubling trick, which also gives the same theoretical bound, but is worse practically. Finally, the authors conduct experiments showing the superior performance of the method over a wide range of hyperparameters.

Strengths: + The paper is well-written, with a careful discussion of the method, the analysis, and prior work. + The result itself is quite nice, and sheds some light on why implicit methods can be expected to outperform standard online mirror descent methods. + The AdaImplicit algorithm updates the learning rates in an adaptive fashion leading to the right regret bounds without any knowledge of the variability, and does so in a practical manner (i.e. without using the doubling trick). + Experiments show that AdaImplicit has good performance and is not sensitive to the tuning of its one parameter, beta.

Weaknesses: - The main weakness as far as I can see is that the same sort of regret bound in terms of the variability of the sequence can be obtained much more easily for the "lazy" version of implicit updates, i.e. a follow-the-regularized-leader style algorithm which computes x_t by minimizing \sum_{s < t} \ell_s(x) + \lambda \Psi(x). This is simply because of the follow-the-leader/be-the-leader inequality. Could the authors comment on this aspect? - One major issue with implicit update methods is that unless there is some special structure, the update requires an "inner loop" of optimization to compute. The authors do mention this point briefly, but a more detailed discussion of this, especially in the context of practical implementations, would be useful. Post rebuttal comments: I agree with the authors that the lazy version of FTRL that I described would be harder to implement in practice. This makes their result more compelling.

Correctness: The claims seem to be correct to the extent I checked the details. I did not verify all the proofs in the appendix though. The experimental methodology is quite sound.

Clarity: The paper is very well-written.

Relation to Prior Work: Comparison with prior work is done well in this paper.

Reproducibility: Yes

Additional Feedback: Post rebuttal: Thank you, I agree that lazy FTRL would be harder to implement in a practical setting, even though it would give the same temporal variability bound. If accepted, please add this point to the paper. Also, some more discussion on implementing the implicit update (as mentioned in the rebuttal) would be useful to have in the next version of the paper.


Review 2

Summary and Contributions: The paper presents algorithms for the online convex optimisation framework with the upper bound formulated in terms of temporal variability, i.e., the sum of maximum differences of \ell_t(x) - \ell_{t-1}(x) over time (I am actually surprised there is no absolute value around the difference in (1)). If temporal variability vanishes, the regret defaults to O(\sqrt{T}). Theorem 6.3 shows optimality of the bound, in the sense that the algorithm of the paper can be made to suffer particular regret.

Strengths: The paper provides an interesting answer to a very natural question.

Weaknesses: Of course, one cannot help asking if the lower bound extends to an arbitrary algorithm. One expects it should...

Correctness: Please confirm there is no absolute value in (1). Is this correct?

Clarity: Yes

Relation to Prior Work: Yes. It would be interesting to know what the result of the paper implies for the changing dependency framework in prediction with expert advice (E. Moroshko, N. Vaits, and K. Crammer. Second-order non-stationary online learning for regression. Journal of Machine Learning Research, 2015; Y. Kalnishkan, An Upper Bound for Aggregating Algorithm for Regression with Changing Dependencies, 2016).

Reproducibility: Yes

Additional Feedback: Thank you for the answer to my comments. I am keeping my score.


Review 3

Summary and Contributions: This paper develops a new analysis framework of online mirror descent using implicit updates, and derives a static regret bound in terms of the variability of loss functions $V_T$. Then a refined version of the implicit algorithm is proposed using adaptive learning rates, which is then proved able to meet the lower bound in settings where “loss functions vary slowly”.

Strengths: This paper considers a new setting that lies between the static setting and the dynamic setting (i.e., the loss functions only vary slowly). Although it is not presented clear enough in this paper, such setting will possibly provide a new vision in this area and may be useful in some specific scenarios.

Weaknesses: This paper does not present its learning setting clear enough. In general, this work seems to be conducted in static setting, as the learning objective is the static regret. However, the bound is derived in terms of the variability of loss functions $V_T$, which is a characteristic in dynamic setting. Throughout the paper the authors claim that this work is more suitable to “small” V_T, but without further explanation or justification. I then wonder what is “small” V_T (this even contradict the experiment setup where “there is no reason to believe the temporal variability is small”). This work has a strong overlap with previous work (FIOL [1]), thus the contribution and novelty is very limited. More specifically, the algorithm seems to be a special case of FIOL which omits the regularization term, and the theoretical analysis is also very similar (see my detailed remarks below). The main novelty in theoretical analysis is to use $V_T$ to bound the gain, which is quite straightforward to deduce. Using adaptive learning rate to improve the algorithm also seems to be normal, as it is directly borrowed from AdaFTRL. [1] C. Song, J. Liu, H. Liu, Y. Jiang, and T. Zhang. Fully implicit online learning.

Correctness: Yes. They are correct.

Clarity: Yes.

Relation to Prior Work: This paper does not discuss its relation with prior works in the dynamic online setting where the variability $V_T$ is also a crucial characteristic to measure how drastic the environment is varying.

Reproducibility: Yes

Additional Feedback: Typo: Line 32, choose to not use -> choose not to use. (update after rebuttal) After reading the response, I understand the setup, and acknowledge the contribution as the first work to link the regret of implicit updates to the variability of losses and further exploit this quantity to attain a smaller regret. However, I still think the overlaps with previous works are somehow strong, which affects the novelty of this paper. Specifically, the designed algorithm is actually a special case of FIOL, and theoretical derivations for the step-wise lemma (Lemma 1) and the regret bound are essentially the same as those in [18]. Given these points, I decide to raise my score to borderline reject.

[Author Response · NeurIPS 2020]

We would like to thank all the reviewers for their detailed feedback, for appreciating our writing style (R1,R2, R3), and
for recognizing that we provide an *interesting answer to a natural question* (R2), also in a *practical manner* (R1).

**REVIEWER 1. Q**: Regret bound in terms of the variability of the sequence can be obtained for FTRL style algorithms.
**A**: This is indeed true, thank you for pointing it out! However, to obtain it you would have to run FTRL with full losses,
rather than the common **linearized** version. Unfortunately, this would entail solving a constrained convex optimization
problem whose size (in terms of number of functions) grows each step, that would be slow even when the implicit
updates have closed form expressions, e.g., linear classification with hinge loss.

**Q**: Unless there is some special structure, the implicit update requires an "inner loop" of optimization to compute.
**A**: It is true that implicit updates in general require a heavier computational burden. However, as illustrated in Appendix
C, there are some very important and practical cases where the updates can be computed in closed form. Also, the
"inner loop" does not seem a serious problem in practice, people do use these kind of algorithms. See for example the
nice posts on Alex Shtof's blog on proximal (aka implicit) updates.

**REVIEWER 2. Q**: Does the lower bound extends to an arbitrary algorithm?
**A**: The lower bound does hold for any deterministic algorithm on constrained domains. For randomized ones, the proof
should be changed and we suspect the lower bound to hold.

**Q**: Please confirm there is no absolute value in (1). Is this correct?
**A**: That's correct, no absolute value in Eq. 1, contrarily to the usual definition of temporal variability in dynamic regret
papers. So, our definition is stronger, since it could lead to a negative term. This is not so surprising: as R1 points out,
FTRL with full losses would depend on a similar quantity at the expense of a growing computational complexity.

**Q**: What does the result of the paper imply for the changing dependency framework in prediction with expert advice?
**A**: We also think it would be interesting to extend our results to the dynamic case. Even if we are not dealing specifically
with the setting of online regression, we believe that it should be possible to get bounds which retain the minimum
between two quantities, one involving $V_T$ and the other in terms of the variation of the comparator sequence (aka path
length), as done in Moroshko et al. [2015] and Kalnishkan [2016].

**REVIEWER 3**. **Q**: This paper considers a new setting that lies between the static setting and the dynamic setting.
**A**: **This is not a new setting**, we are dealing with the **static** setting, as specified in the abstract and in the introduction
(see lines 5; 19–25). The reviewer seems to imply that the dynamic setting is the only one where losses vary slowly, but
this is incorrect: it only differs from the static setting for the choice of a set of competitors rather than a single one. The
confusion of the reviewer seems to stem from the fact that no prior analysis of implicit updates had a term that depends
on the variability of the losses. Shedding light on this novel aspect of implicit updates is our main contribution.

**Q**: This paper does not discuss its relation with prior works in the dynamic online setting.
**A**: This seems factually incorrect: We do have a discussion with prior work for dynamic environments: please see lines
60–64, 71–74, and related references.

**Q**: The authors claim that this work is more suitable to "small" $V_T$, but without further explanation or justification.
**A**: This seems incorrect: we **never** claim that our algorithm is suitable for small $V_T$. Instead, our bound is a minimum
between two quantities. In the worst case our algorithm recovers the $\sqrt{T}$ bound, but in other situations where $V_T$ is
small it adapts to it (contrarily to standard algorithms like linearized MD or FTRL) and can have a better bound. In
other words, we have a classic "best of both worlds" bound.

**Q**: This work has a strong overlap FIOL [31], thus the contribution and novelty is very limited.
**A**: Our paper has not only overlaps with FIOL, but also with [18] and [20]. All of them provide an intuition that it's
possible to get a gain from the analysis, *but they fail to quantify it in the final bound*. We also **clearly** point out this
overlap, see lines 125–127. In particular, from the analysis in FIOL we can see a potential gain in the bound in their Eqs.
(6–7). On the other hand, it is not clear how their negative term in the final bound could lead to something which is less
than $\sqrt{T}$. Also, any potential gain is entirely destroyed by their learning rate $\eta_t \propto 1/\sqrt{t}$ which leads to a $\sqrt{T}$ bound.

**Q**: The main novelty is to use $V_T$ to bound the gain, which is quite straightforward to deduce.
**A**: We respectfully disagree: almost anything in Science is straightforward after somebody points it out. Yet, none of the
previous work pointed out the connection between implicit updates and $V_T$. Unless we missed other related work, this
is the first work where implicit updates could have a quantifiable advantage (i.e., possibly O(1) regret) over OMD ones.

**Q**: I then wonder what is "small" $V_T$ (this even contradict the experiment setup where [...])
**A**: We do show in our synthetic experiment a situation when $V_T$ is small and our algorithm is much better compared to
the other baselines. On standard real-world datasets, we say that there is no reason to believe that $V_T$ is small, but we
want to show that our algorithm is still competitive. On a side note, we are not aware of any paper on dynamic regret
with experiments on real world datasets with small $V_T$.

[Meta-Review · NeurIPS 2020]

This paper considers the implicit update algorithm for online learning (a.k.a. proximal updates in the optimization literature). It is shown that the algorithm achieves a regret bound that is adapted to the variability of the sequence of loss functions. This holds even without the smoothness of the loss. I believe this is a firm contribution to the fields of online learning and stochastic optimization. Firstly, Implicit updates are known to have practical advantages, but their theoretical understanding has been limited to the fact that they enjoy the same worst-case regret guarantees as their explicit counterparts. This is one of a very few works (if not the first one) which shows a nontrivial advantages of the implicit methods and thus makes a significant progress in better understanding of their behavior. Secondly, the previous approaches to dealing with limited temporal variability of the loss, such as optimistic MD, require the loss to be linear or smooth. In contrast, the authors show that their method achieves O(1) regret when the temporal variability is constant without smoothness assumption. Finally, the authors provide an adaptive version of the algorithm (AdaImplicit) which does not need any knowledge on the variability V_T, and simultaneously guarantees a regret bound adapted to the variability and the worst-case bound of AdaFTRL. The was a discrepancy in the scores assigned by the reviewers. The paper was found to be well-written and interesting by two of the reviewers who also appreciated the novel results. One of the reviewers found the paper to be a rather incremental update of the prior work on FIOL, and the usage of temporal variability to be inappropriate in the static regret framework. In my own opinion, these doubts were properly addressed in the rebuttal.